# Marker Identification of the Grade of Dysplasia of Intraductal Papillary Mucinous Neoplasm in Pancreatic Cyst Fluid by Quantitative Proteomic Profiling

**DOI:** 10.3390/cancers12092383

**Published:** 2020-08-23

**Authors:** Misol Do, Hongbeom Kim, Dongyoon Shin, Joonho Park, Haeryoung Kim, Youngmin Han, Jin-Young Jang, Youngsoo Kim

**Affiliations:** 1Department of Biomedical Sciences, Seoul National University College of Medicine, Seoul 03080, Korea; jm97129@naver.com (M.D.); sdymath777@naver.com (D.S.); 2Department of Surgery, Seoul National University College of Medicine, Seoul 03080, Korea; surgeonkhb@gmail.com (H.K.); vickijoa@naver.com (Y.H.); 3Department of Biomedical Engineering, Seoul National University College of Medicine, Seoul 03080, Korea; tryjpark@gmail.com; 4Department of Pathology, Seoul National University College of Medicine, Seoul 03080, Korea; haeryoung.kim@snu.ac.kr

**Keywords:** pancreatic cyst fluid, intraductal papillary mucinous neoplasm (IPMN), mucinous cystic neoplasm (MCN), serous cystic neoplasm (SCN), biomarkers, LC-MS/MS

## Abstract

The incidence of patients with pancreatic cystic lesions, particularly intraductal papillary mucinous neoplasm (IPMN), is increasing. Current guidelines, which primarily consider radiological features and laboratory data, have had limited success in predicting malignant IPMN. The lack of a definitive diagnostic method has led to low-risk IPMN patients undergoing unnecessary surgeries. To address this issue, we discovered IPMN marker candidates by analyzing pancreatic cystic fluid by mass spectrometry. A total of 30 cyst fluid samples, comprising IPMN dysplasia and other cystic lesions, were evaluated. Mucus was removed by brief sonication, and the resulting supernatant was subjected to filter-aided sample preparation and high-pH peptide fractionation. Subsequently, the samples were analyzed by LC-MS/MS. Using several bioinformatics tools, such as gene ontology and ingenuity pathway analysis, we detailed IPMNs at the molecular level. Among the 5834 proteins identified in our dataset, 364 proteins were differentially expressed between IPMN dysplasia. The 19 final candidates consistently increased or decreased with greater IPMN malignancy. CD55 was validated in an independent cohort by ELISA, Western blot, and IHC, and the results were consistent with the MS data. In summary, we have determined the characteristics of pancreatic cyst fluid proteins and discovered potential biomarkers for IPMN dysplasia.

## 1. Introduction

The incidental detection of pancreatic cystic lesions (PCLs) has increased in recent years due to the implementation of various screening methods and the advancement of medical imaging technologies, such as magnetic resonance imaging (MRI), computed tomography (CT), and endoscopic ultrasound (EUS) [1,2,3,4,5]. In response, many studies have attempted to develop screening methods that aid in the therapeutic decision-making with regard to PCLs, including intraductal papillary mucinous neoplasm (IPMN), which has been detected most frequently as a precursor lesion of pancreatic cancer [5].

IPMN stages vary significantly as the malignancy progresses from benign to malignant–low-grade dysplasia (LGD), high-grade dysplasia (HGD), and invasive IPMN [6]. LGD is considered primarily to be amenable to active surveillance, whereas the lesions in HGD and invasive IPMN require surgical intervention [7,8], necessitating the accurate classification of cystic lesions for appropriate patient management. Currently, 3 guidelines are used widely for establishing the treatment strategy for IPMN patients [9,10,11]. However, the standard for determining whether to conduct active surveillance or surgical intervention and the diagnostic accuracy in determining IPMN grade differ between guidelines. Thus, the same patient can be treated differently, depending on which guideline is followed by the clinician. Consequently, establishing a treatment strategy that is based solely on these guidelines is problematic in actual clinical practice [12].

To examine the discrepancy between these guidelines, nomograms have been developed to predict low-risk and high-risk IPMNs [13,14]. However, the nomogram-derived objective risk score has limited diagnostic accuracy. The following factors are used to calculate the risk score: (1) Abdominal imaging, such as MRI, CT, and EUS; (2) carcinoembryonic antigen (CEA) and carbohydrate antigen 19-9 (CA19-9) levels; (3) analysis of KRAS and GNAS mutations; and (4) cyst fluid cytology. MRI and CT scans are inconsistent in differentiating between cyst types, as evidenced by their wide range in diagnostic accuracy (20% to 80%) [15]. EUS also suffers from poor accuracy (51% to 59%) and cannot distinguish benign cystic lesions from malignant cysts [16]. The most extensively studied biomarker, CEA, has low accuracy (60% to 80%) when used to discriminate between mucinous and nonmucinous cystic lesions [17]. Another tumor marker, CA19-9, has a specificity of 81%, which is offset by its low sensitivity (15%) in differentiating mucinous and nonmucinous cystic lesions [18]. Analysis of KRAS mutations has 100% specificity but is not sufficiently sensitive (50%) to determine IPMN dysplasia [19]. Similarly, although analyzing GNAS mutations is adequate for distinguishing IPMNs from other types of pancreatic cysts, they cannot predict the grade of dysplasia, because they generally occur early in IPMN development [20]. Cyst fluid cytology for mucinous cysts has low diagnostic accuracy (54% sensitivity and 93% specificity) [21]. Further, cytological examination is only applicable to cases in which there is a sufficient concentration of observable cells in the cyst fluid [22]. Thus, it is likely that patients have undergone unnecessary surgical interventions due to the absence of an accurate method for determining the malignancy of IPMN [23], necessitating novel biomarkers that improve the accuracy of the diagnosis of IPMN.

Pancreatic cyst fluid has several advantages over serum and plasma with regard to the discovery of markers for IPMN [24,25,26]. Because cyst fluid is composed of secreted proteins from surrounding tumor cells, the concentration of potential biomarkers in cyst fluid is approximately 1000 times higher than in blood. In addition, these candidates closely represent changes in the dysplastic epithelium [27], and cyst fluid can be collected by endoscopic ultrasound-guided fine needle aspiration (EUS-FNA), a safe and minimally invasive method [28].

Mass spectrometry (MS)-based proteomic approaches are being increasingly applied to identify markers that are related to specific diseases, based on their ability to screen thousands of proteins simultaneously to obtain hundreds of differentially expressed proteins (DEPs) in small amounts of samples [29,30]. For instance, a study by Jabbar et al. concluded that quantifying CEA by a conventional method requires 1000 times more cyst fluid than MS analysis [31]. Thus, an MS-based approach is the most suitable platform for screening biomarkers in cyst fluid.

Existing diagnostic modalities (CT, MRI, EUS, CEA, and CA19-9 levels; KRAS and GNAS mutations; and cyst fluid cytology) are insufficient for accurately classifying IPMN patients due to their low diagnostic accuracy [15,16,17,18,19,20,21]. Consequently, many reports, including proteomic studies, have examined methods of discovering biomarkers to improve the diagnostic accuracy for PCLs. In our previous study, we identified potential biomarkers of the histological grades of IPMNs using cyst fluid that was obtained exclusively from IPMN patients by LC-MS/MS [32]. In the current study, we aimed to discover marker candidates for IPMN dysplasia from an expanded cohort that included IPMNs and other PCLs (mucinous cystic neoplasm (MCN) and serous cystic neoplasm (SCN)) by mass spectrometry to better reflect actual clinical circumstances to help classify various PCLs and avoid unnecessary pancreatic resection for low-risk IPMN patients.

## 2. Results

### 2.1. Results of Proteomic and Bioinformatic Analyses

#### 2.1.1. In-Depth Quantitative Proteomics of Pancreatic Cyst Fluid

A mass spectrometry-based method, based on our previous study, was used to analyze a cohort of cyst fluid samples to measure the changes in protein expression with respect to the progression of IPMN [32]. The overall procedure for discovering markers of IPMN progression, from sample preparation to the LC-MS/MS analysis, is depicted in Figure 1. The discovery cohort included 30 pancreatic cyst fluid samples from 3 types of IPMN (LGD, HGD, and invasive IPMN) and other PCLs (MCN and SCN). The pooled samples were fractionated and analyzed in parallel to generate a peptide library, which was used to expand the coverage of identified proteins for individual samples. Each fractionated sample was analyzed once, whereas all individual samples were analyzed in triplicate on a Q Exactive mass spectrometer.

Raw MS data were processed in MaxQuant (version 1.6.0.16), and the statistical analysis was performed with Perseus (version 1.6.1.1). The MaxQuant analysis identified 1,314,934 spectral matches, 56,583 peptides, and 5834 protein groups, 5774 of which were quantifiable (Appendix A). For label-free quantification, 5578 and 3249 proteins were identified in the peptide library and 30 individual cyst fluid samples, respectively. A total of 2993 proteins (92.1%) that were identified in individual samples overlapped with the peptide library (Figure 2A). The quantified proteins in individual cyst fluids accounted for 86.5% of the 3249 identified proteins (Figure 2B). Notably, the 3 IPMN groups had approximately twice the number of quantified proteins (2220–2500) than MCN (1218) and SCN (1346) (Appendix A). Overall, the number of quantified proteins varied significantly, even within histological groups. Specifically, the identified and quantified proteins in each sample ranged from a minimum of 657 identified (298 quantified) in LGD 10 to a maximum of 2587 identified (2014 quantified) in invasive IPMN 1 (Appendix A).

To improve the proteome coverage, the “match between runs” feature in MaxQuant was utilized to align the retention times and MS/MS spectra of the individual sample against the peptide library [33]. In total, an additional 773 and 420 proteins were identified and quantified, respectively, across all individual samples. LGD 6 showed the largest increase in the number of identified and quantified proteins–by 457 and 235, respectively. On average, 100 more peptides were identified in each sample (Figure 2C, Appendix A). This result demonstrates that the overall proteome coverage of individual cyst fluid samples rose, thereby enlarging the pool of potential biomarker candidates.

The dynamic range of protein expression levels spanned over 7 orders of magnitude, but most proteins (95%) were expressed within 4 orders of magnitude (Appendix A). Of these proteins, the levels of pancreatic cancer-associated proteins, such as MUC5AC, MUC2, and CEA, were high. Five proteins (PNLIP, CPA1, CPB1, PRSS1, and PRSS2) in a smaller dynamic range, as shown in Appendix A, are known to be significantly expressed in the pancreas compared with other organs [34]. In addition, with the exception of PRSS1, these proteins, denoted in blue, are generally exclusive to the pancreas, per Wilhelm [34]. This result confirms the presence of pancreas-specific proteins in our proteome data.

#### 2.1.2. Reproducibility of Data and Comparison with Other Proteome Databases

To evaluate the reproducibility between the technical replicates, the coefficient of variation (CV) values and Pearson correlation coefficients of LFQ intensity values between technical replicates were calculated (Appendix A). The low median CV values (<20%) and high Pearson correlation coefficient (>0.9) between technical replicates indicated that the label-free quantification of cyst fluid samples was highly reproducible.

To examine the composition of pancreatic cyst fluid proteins, our data were compared with various proteome databases and data from past studies. These datasets included the following: (1) SecretomeP, SignalP, and TMHMM; (2) the Human Plasma Protein Database; (3) the Human Protein Atlas; (4) the “core” proteome, which referred to the proteins that were common to 5 major proteome databases, in Wilhelm et al. [34]; and (5) our previous study [32] (Appendix A). These comparative analyses supported that the proteins that were identified in this study had the appropriate characteristics of pancreatic cyst fluid and exceeded the proteome coverage of past cyst fluid proteomes. The expression patterns of 8 marker candidates from our previous report were replicated in the present study but were statistically insignificant (Appendix A). Results on the reproducibility of the data and the comparative analysis with other proteome databases are detailed in Appendix A.

### 2.2. Discoverying Biomarker Candidates of IPMN Dysplasia

#### 2.2.1. Differentially Expressed Proteins between IPMN Dysplasia

The diagram in Appendix A details the discovery of potential markers of IPMN dysplasia. Of the 5834 identified proteins, 2809 were quantifiable in individual cyst fluid samples and had LFQ intensity values in at least 2 technical replicates in 1 biological replicate. Of the 2809 quantified proteins, 1019 had more than 70% measurable LFQ intensities in at least 1 histological group and were deemed usable for the statistical analysis. This criterion was established to ensure that a putative marker candidate represented at least 1 histological group.

To identify DEPs, student’s t-test (*p* < 0.05) was performed for each comparative pair (comparisons 1 to 7). The results, including the statistical significance, p-values, and fold-changes, are detailed in Appendix A. In comparisons 1 (LGD versus HGD), 2 (HGD versus invasive IPMN), and 3 (LGD versus invasive IPMN), 216, 84, and 247 proteins were differentially expressed, respectively–of which 164, 61, and 192 were upregulated.

The variance in expression between comparisons 1 to 3 was depicted in volcano plots. The highlighted final marker candidates including the validation target, CD55, underwent significantly large fold-changes (Figure 3). In the statistical analysis of comparisons 1 to 3, 364 DEPs remained after overlapping proteins were removed from each comparative group (Appendix A). Of the 364 DEPs, 261 were exclusively upregulated, and 80 proteins were exclusively downregulated. The remaining 23 proteins did not have consistent expression patterns across the 3 comparisons (Appendix A).

Gene ontology (GO), KEGG pathway analysis, and Ingenuity Pathway Analysis (IPA) were performed to characterize the 364 DEPs. The GO terms and KEGG pathways were associated with pancreatic cancer and cyst fluid (Appendix A). A total of 216 DEPs from comparison 1 (LGD vs. HGD) and 247 DEPs from comparison 3 (LGD vs. invasive IPMN) were analyzed by IPA. The biological functions that were related to “malignancy” and “molecular secretion” were associated with the DEPs in the core analysis (Appendix A). In addition, in the comparative analysis, pancreas-specific diseases and biological functions that were related to malignancy were confirmed to be enriched to a greater extent in comparison 3 than comparison 1 (Appendix A). All bioinformatics analyses results are detailed in Appendix A.

#### 2.2.2. Biomarker Candidates of IPMN Dysplasia

A total of 364 DEPs passed the statistical analysis (Appendix A). Following the rationale that proteins with significant differences in expression in more comparisons are more likely to be biomarkers, 179 DEPs were designated as initial marker candidates, because they were present in at least 2 of 3 comparative pairs (1 to 3) [32,33]. Subsequently, 27 DEPs had expression patterns that consistently increased or decreased with greater IPMN malignancy. Of them, 13 DEPs that were preferentially expressed in invasive IPMN were statistically significant in comparisons 6 (SCN versus invasive IPMN) and 7 (MCN versus invasive IPMN). Similarly, the remaining 14 DEPs were expressed in LGD and showed significant differences in comparisons 4 (SCN versus LGD) and 5 (MCN versus LGD). Based on the rationale that tumor-associated proteins are generally secreted from surrounding tumor cells, the 19 DEPs predicted to be secreted by SecretomeP, SignalP, and TMHMM were selected as the final marker candidates [35,36].

The heat map in Figure 4 provides an overview of the expression of the 19 final marker candidates of IPMN dysplasia: 7 invasive IPMN-specific marker candidates and 12 LGD-specific marker candidates based on their expression patterns. DEFA3, MUC13, CD55, CPS1, RAB11B, HEXA, and SOD2 were highly expressed in invasive IPMN and expressed at statistically lower levels in other PCLs. In contrast, LEFTY1, AMY2A, KLK1, RNASE1, CELA2A, CELA3A, CPA1, CPB1, CEL, AMY2B, GP2, and CTRC were highly expressed in LGD and but significantly lower in other cystic lesions (Figure 5, Appendix A).

Appendix A details the results of the statistical analysis of the 19 potential markers (7 upregulated proteins and 12 downregulated proteins), including statistical significance, p-values, and fold-changes for each comparative group. A total of 16 proteins, with the exception of RAB11B, KLK1, and CELA2A, were pancreatic tissue-specific proteins, according to the Human Protein Atlas. In addition, 15 proteins, with the exception of DEFA3, MUC13, RAB11B, and LEFTY1, were observed in plasma or serum, per the Plasma Proteome Database (PPD) (Appendix A).

The fold-changes of the 19 final marker candidates were assessed in relation to the general distribution of the 1019 proteins that were used for the statistical analysis. To this end, potential markers from comparisons 1 and 3 were displayed in a dynamic range and ordered, based on their fold-change. (Figure 6A,B). Excluding CD55, RAB11B, and CPS1 in comparison 1, 16 proteins had p-values below 0.05 and lay generally near the 2 extremes of the dynamic range. Similarly, in comparison 3, 19 candidates were statistically significant (*p* < 0.05) and located near the upper and lower extremes of the dynamic range. All potential markers had higher fold-changes than CEA, the most well-established pancreatic cancer-associated marker.

Upstream regulator analysis in IPA was conducted to predict the upstream proteins of the final candidates and their biological functions. This analysis predicted the top 20 likely upstream regulators that modulate the 5 final candidates (MUC13, CD55, CPS1, SOD2, and LEFTY1). A total of 11 regulator proteins were associated with SOD2, 4 proteins correlated with CD55, 4 proteins were linked to LEFTY1, and 1 was associated with CPS1 and MUC13 (Figure 6C, Appendix A). The upstream regulators were the following molecular types: Kinase, enzyme, g-protein-coupled receptor, transcription regulator, transporter, and ion channel. The upstream regulators were tumor suppressors, pancreatic mitogens, and other key factors of cancer progression, according to several publications, supporting the credibility of the marker candidates [37,38,39,40,41,42,43,44,45,46,47,48,49,50,51].

### 2.3. Determination of CD55 Levels by Antibody-Based Methods

#### 2.3.1. Determination of CD55 by Enzyme-Linked Immunosorbent Assay (ELISA) and Western Blot

The selection of the validation target was based on a protein’s predominance in LGD or invasive IPMN and its statistical significance (Appendix A). Accordingly, CD55 was chosen as a validation target of IPMN dysplasia for 3 main reasons: (1) It was highly expressed in all (100%, 5/5) invasive IPMN samples (Figure 4), (2) it had the smallest p-value of all potential markers from all comparative groups that involved invasive IPMN (Figure 5, Appendix A), and (3) it had the highest fold-change between comparisons 2 (HGD vs. invasive IPMN) and 3 (LGD vs. invasive IPMN) (Figure 3).

CD55 was validated in 70 cyst fluid samples (22 LGD, 5 HGD, 14 invasive IPMN, 13 MCN, and 16 SCN) by ELISA. The demographics and clinical information (cyst characteristics and CEA and CA19-9 levels) are described in Appendix A. CD55 concentrations in individual cyst fluid samples were calculated and demonstrated in 2 types of IPMN classification (Figure 7). The concentration of CD55 was the highest in invasive IPMN, and its expression patterns generally correlated with LFQ intensity values. CD55 concentrations in invasive IPMN (mean: 1.354 ng/mL, STDEV: 1.532 ng/mL) were significantly higher than in LGD (mean: 0.598 ng/mL, STDEV: 1.045) (*p* < 0.05). In addition, CD55 concentrations in high-risk IPMN (mean: 1.219 ng/mL, STDEV: 1.567) were significantly higher versus low-risk IPMN (mean: 0.598 ng/mL, STDEV: 1.045). The expression levels of CD55 in invasive IPMN compared with SCN and MCN were statistically significant (*p* < 0.05). The intraplate and interplate repeatability of the CD55 ELISA was evaluated by measuring 3 replicates of a total of 21 positive and negative control samples. The CV values for the intraplate and interplate repeatability were less than 20% (Appendix A). A detailed description of the precision of the CD55 ELISA can be found in the Appendix A.

To reconfirm CD55 expression between the 5 cystic lesions, Western blot was conducted using 30 cyst fluid samples (8 LGD, 4 HGD, 8 invasive IPMN, 5 MCN, and 5 SCN). Ponceau S staining was included as a loading control to confirm that comparable amounts of individual samples were loaded onto each gel. The resulting CV value was 12.53% (Appendix A). The signal intensity of CD55 was the highest in invasive IPMN, and its expression patterns correlated with the MS analysis findings (Appendix A). The signal intensities in invasive IPMN were significantly higher than in LGD (*p* < 0.001), MCN (*p* < 0.01), and SCN (*p* < 0.01) (Appendix A). In addition, the signal intensities in high-risk IPMN were significantly higher versus low-risk IPMN (*p* < 0.001) (Appendix A).

#### 2.3.2. Immunohistochemistry (IHC) of CD55 and Myeloperoxidase (MPO)

Immunohistochemical stains for CD55 and MPO were performed on formalin-fixed paraffin-embedded (FFPE) tissue sections from SCN, LGD, HGD, and invasive IPMN (Appendix A). CD55 expression was observed predominantly in the apical border of the tumor epithelial cells, showing increased distribution and intensity, in accordance with the histological grades of IPMN, whereas strong membranous staining was observed in invasive IPMN (Appendix A). Also, we examined MPO, a neutrophil marker, to identify neutrophil infiltration in invasive IPMN, based on a previous study that reported that CD55 is responsible for transepithelial migration of neutrophils [52]. As expected, neutrophil infiltration increased from LGD to HGD and invasive IPMN, which had the highest neutrophil counts (Appendix A). CD55 expression and neutrophil infiltration were not observed in SCN.

## 3. Discussion

In this study, we discovered reliable marker candidates of IPMN dysplasia using cyst fluid from IPMN, MCN, and SCN patients by LC-MS/MS and investigated their molecular characterization, based on the advantages of pancreatic cyst fluid and MS-based proteomic approaches [27,28,29,30]. The current diagnostic screens cannot accurately determine the IPMN-associated grade of dysplasia [1,4], leading to unnecessary surgical resections for low-risk IPMN patients [23]. In addition, most studies have focused on discovering diagnostic markers that differentiate IPMNs from other PCLs, rather than the grade of dysplasia in IPMN [53]. Our report is the first study to discover potential markers of IPMN dysplasia using cyst fluid from 3 major types of PCL by LC-MS/MS and validated them by orthogonal method.

Our proteome data have 3 notable aspects: (1) increased depth of proteome coverage through the use of a peptide library, (2) high reproducibility, and (3) an abundance of pancreas-associated proteins. We hypothesized that a high proportion of proteins are coexpressed in cyst fluid and pancreatic cancer cell lines. Thus, we expected that the “match between runs” feature in MaxQuant would help identify proteins in cyst fluid that are normally unidentifiable without a peptide library [54]. Consistent with our expectation, approximately twice as many proteins were identified in this dataset than in our previous study (Appendix A), constituting the largest proteomic dataset of pancreatic cyst fluid [32,55,56]. Consequently, we discovered potential markers of the histological grades of IPMN in a larger pool of proteins. Further, the low median CV values and high Pearson correlation coefficients of LFQ intensities between technical replicates indicated that the individual samples were injected into the mass spectrometer without significant variance and that the technical replicates were analyzed reproducibly. (Appendix A). In previous studies, it was concluded that pancreatic cyst fluid contains secreted proteins from surrounding tumor cells [35,36,53] and plasma proteins that penetrate into the cyst epithelium due to tissue injury or the enhanced permeability and retention (EPR) effect of the surrounding blood vessels [57]. Tumor-promoting mediators, such as immunosuppressive cytokines, that are released from cancer-associated fibroblasts and mast cells in the tumor microenvironment can promote the neoplastic evolution of IPMN [58,59,60]. They induce an aggressive phenotype and drug resistance in premalignant pancreatic lesions. One possible mechanism of the malignant evolution of IPMN, in light of our findings and existing literature, is that the expression of CD55, promoted by the immunosuppressive cytokine, interleukin-4, prevents complement-dependent cytotoxicity in cancer cells, which consequently accelerates the malignant transformation of IPMN dysplasia [61,62]. The results of the comparative analyses with the 3 databases for the secretome analysis and the Human Plasma Protein Database (Appendix A) support these findings [35,36,53]. In addition, 15 of the final marker candidates were detected in plasma and serum, supporting their viability in a blood-based assay [63,64]. A high proportion (approximately 90%) of pancreas-associated proteins were identified in our dataset (Appendix A) and 5 pancreas tissue-specific proteins, as defined by Wilhelm et al. [34], were in the top 25 proteins (Appendix A), demonstrating that our proteome has sufficient coverage of pancreas-specific proteins.

GO and KEGG pathway analyses were performed to identify key processes of the 364 DEPs in IPMN dysplasia. The results indicated the enrichment of terms that pertained to tumorigenesis: pancreatic secretion, enzymatic activity, and malignancy (Appendix A). GO terms that were related to “pancreatic secretion” and “molecular transport” were highly ranked in the GO and KEGG analyses, suggesting that the DEPs in IPMN dysplasia are generally secreted from the surrounding tumor cells. One of the most highly enriched GO terms, “proteolysis,” is the most fundamental feature of malignancy [65,66]. Proteolytic degradation of ECM constituents accelerates tumor cell growth, migration, and angiogenesis. The most highly enriched KEGG pathway, “complement and coagulation cascades,” is associated with tumor growth and metastasis [67,68]. Complement activation promotes an immunosuppressive microenvironment and thus induces angiogenesis, activating cancer-related signaling pathways. In addition, several studies have reported increases in complement activity in biological fluids from cancer patients [69,70]. Coagulation cascades can be activated directly by cancer procoagulants, which are released by tumor cells.

The criteria for discovering marker candidates of IPMN dysplasia that comprised 8 steps (Appendix A) and the association of the final marker candidates with the pancreas-related disease (IPMN and PDAC) and malignancy support the credibility of the potential markers. For instance, MUC13 has been studied extensively. According to 2 previous studies with similar goals as ours, MUC13 increased with histological grade of IPMN [71,72]. In addition, several studies have indicated that MUC13 is highly upregulated in PDAC tissue but not in adjacent normal tissue and is related to PDAC progression [73,74,75]. CD55 is involved in the dedifferentiation, invasiveness, migration, and metastasis of tumors and association with pancreatic cancer [76,77]. Further, Iacobuzio-Donahue confirmed that CD55 is highly expressed in pancreatic cancer when measured by microarrays [78]. Two previous studies that aimed to discover protein markers of mucinous and nonmucinous cysts selected AMY2A as a biomarker, consistent with the expression patterns in our study [26,55]. In addition, AMY2A was expressed at higher levels in nontumor versus PDAC tissues [79]. According to the label-free quantification data of another group, CPB1, a member of the carboxypeptidase family, was confirmed to be downregulated in PDAC tissue [80]. The biological functions and expression patterns of our potential markers are consistent with previous studies.

Our data also showed that most of the 20 upstream regulators that modulate 5 of the potential markers were tumor suppressors, pancreatic mitogens, and angiogenesis-related molecules, which are significantly related to pancreatic cancer according to previous publications (Figure 6C, Appendix A) [37,38,39,40,41,42,43,44,45,46,47,48,49,50,51]. The association between upstream proteins of the potential markers and cancer progression increases the credibility of our candidates. For instance, it is plausible that the downregulation of 3 upstream regulators of SOD2 (IGFBP7, KL, BTG2), which are potential tumor suppressors, leads to an increase in SOD2 when the malignancy of IPMN worsens.

Of the final marker candidates, CD55 was validated using 70 individual cyst fluid samples by ELISA as it evidently differed in expression between the histological groups of IPMN. In specific, CD55 had the lowest p-values between all comparative groups and the highest fold-changes in comparisons 2 (HGD versus invasive IPMN) and 3 (LGD versus invasive IPMN) (Figure 3, Figure 5, Appendix A). Although previous studies have concluded that CD55 is associated with the dedifferentiation and invasiveness of tumors [76,77], this study is the first to report CD55 as a marker of IPMN dysplasia. The statistical significance in the ELISA analysis was lower compared with the MS analysis. Further, the CD55 concentrations by ELISA were generally lower in all sample groups (Figure 7). However, the low statistical power and concentrations do not diminish the value of this potential marker because the expression patterns of CD55 by ELISA were consistent with our MS data. In addition, the Western blot and IHC results for CD55 were consistent with the MS expression data, further supporting CD55 as a potential marker of IPMN dysplasia (Appendix A).

Several studies have examined CD55 as a potential biomarker and therapeutic target, establishing that CD55 is frequently upregulated in various cancer types and can serve as an indicator of cancer progression [81]. In a preclinical study, Saygin et al. demonstrated that CD55 maintains self-renewal and cisplatin resistance in endometrioid tumors to accelerate tumor development [82]. Other preclinical studies concluded that silencing CD55 enhances the therapeutic efficacy of rituximab [83] and the anti-HER2 monoclonal antibodies trastuzumab and pertuzumab [84]. In this context, it is evident that CD55 has a significant function in tumor progression, based on past studies and our findings. When considering the aforementioned preclinical studies regarding CD55, this marker has a significant potential as a reflection of the neoplastic evolution of IPMN.

In our previous study, which applied a similar MS-based approach, we reported potential markers that can differentiate the histological grades of IPMN using only cyst fluid from IPMN patients [32]. However, the exclusion of other PCLs, such as MCNs and SCNs, is not applicable to an actual clinical environment. To compensate for this limitation, cyst fluid from other PCLs (MCN and SCN) as well as IPMN were included to increase the likelihood of discovering more clinically relevant marker candidates. Further, we increased the size of our validation cohort by 4-fold and measured protein levels by ELISA instead of Western blot, due to the higher sensitivity and specificity of the former. In contrast to Western blot, which cannot distinguish proteins with similar molecular weights, ELISA is highly specific to its target epitope and consequently generates credible quantitative concentrations [85].

However, several challenges remain to be addressed. Our study was limited by its simplistic design (single-center). Using a larger cohort from multiple centers would decrease the potential bias that might have been unique to the cohort in this study. Thus, it would be necessary to examine the clinical potential of CD55 through randomized, controlled, multicenter validation in a large cohort. Although 5 marker candidates, including CD55, were statistically significant (*p* < 0.05) in the univariate analysis for predicting the neoplastic evolution of IPMN, no significant covariates were identified in the multivariate analysis. One explanation is the relatively small sample size used in the study, which comprised 10 low-risk IPMN and 10 high-risk IPMN samples. Thus, further validation with more samples is necessary for obtaining more reliable results. Another limitation is that all of the cyst fluid in this study was collected from the tissue of patients who were undergoing resection rather than EUS-guided aspiration, which is a closer representation of clinical practice. Thus, the evaluation of CD55 using preoperative EUS-guided cyst fluid is a natural next step in validating the diagnostic value of CD55 as a marker for IPMN dysplasia.

## 4. Materials and Methods

### 4.1. Patients and Cyst Fluid Samples and Preparation

Cystic fluid samples were collected from 30 patient specimens (20 IPMN, 5 MCN, and 5 SCN) immediately after pancreatectomy at Seoul National University Hospital between April 2013 and June 2017. IPMN samples were classified as low-grade dysplasia (LGD, n = 10), high-grade dysplasia (HGD, n = 5), and invasive IPMN (n = 5). The same samples were also categorized as low-risk IPMN (LGD) and high risk-IPMN (HGD and invasive IPMN). The patient data and characteristics of the cystic lesions are summarized in Table 1 and detailed in Appendix A. At least 200 µL of cyst fluid was aspirated from patients to acquire sufficient protein for analysis. The aspirated cyst fluid samples were stored at −80 °C until sample preparation. All contents of this research were approved by the Institutional Review Board (IRB No. 1304-121-486), and all participants provided written informed consent.

### 4.2. Pancreatic Cyst Fluid Sample Preparation

Brief sonication was performed to remove the mucus in a 1.5-mL Eppendorf tube. The sonicated samples were then centrifuged (15,000 rpm, 20 min, 4 °C) to obtain a supernatant [32]. After measuring protein concentration, equal amounts of protein in each sample were precipitated with cold acetone. The proteins were denatured using SDT lysis buffer (4% SDS, 0.1 M DTT, 0.1 M Tris-Cl, pH 7.4). The sample preparation comprised tryptic digestion with filter-aided sample preparation (FASP) and desalting with homemade StageTips [86,87]. StageTip-based, high-pH peptide fractionation was performed only for pooled samples that were used to generate the peptide library. The sample preparation is detailed in Appendix A.

### 4.3. LC-MS/MS and Statistical Analysis

Liquid chromatography-tandem mass spectrometry (LC-MS/MS) analysis was performed on a Q Exactive mass spectrometer that was equipped with an EASY-Spray ion source (Thermo Fisher Scientific, Waltham, MA, USA), coupled to an Easy-nano LC 1000 (Thermo Fisher Scientific, Waltham, MA, USA) [32,88]. All raw MS files were processed in MaxQuant, version 1.6.0.16 [89] with the built-in Andromeda search engine [90] against the Uniprot human database (88,717 entries; version from December 2014). All generated proteomic data have been submitted to the ProteomeXchange Consortium (http://proteomecentral.proteomexchange.org/) via the PRIDE partner repository, with PXD016127 as the identifier [91,92]. The LC-MS/MS analysis and raw data search are detailed in Appendix A.

The statistical analysis was performed in Perseus (version 1.6.1.1) per our previous studies [32,93]. Student’s *t*-test (*p* < 0.05) was applied to find significantly changed proteins. The 7 comparative pairs that were subjected to statistical analysis were as follows: LGD versus HGD (comparison 1), HGD versus invasive IPMN (comparison 2), LGD versus invasive IPMN (comparison 3), SCN versus LGD (comparison 4), MCN versus LGD (comparison 5), SCN versus invasive IPMN (comparison 6), and MCN versus invasive IPMN (comparison 7). The statistical analysis is detailed in Methods.

### 4.4. Bioinformatics Analysis

The gene ontologies (GOs) of the analyzed DEPs were explicated with the DAVID bioinformatics tool (http://david.abcc.ncifcrif.gov/) and UniprotKB database (http://www.uniprot.org/). Pathway analysis was performed using the KEGG database (http://www.genome.jp/kegg/). Putative secretory proteins were confirmed using SignalP 4.1 (http://www.cbs.dtu.dk/services/SignalP/), SecretomeP 2.0 (http://www.cbs.dtu.dk/services/SecretomeP/), and TMHMM, server 2.0 (http://www.cbs.dtu.dk/services/TMHMM/) [94,95,96]. The Plasma Proteome Database (PPD) was used to estimate the percentage of proteins that were identified simultaneously in human plasma and in this study [63,64]. The proteins that were identified in this study were crossreferenced with mRNA and protein expression in the “pancreatic category” of the Human Protein Atlas (http://www.proteinatlas.org/). Ingenuity Pathway Analysis (IPA) was used for the functional analysis (Ingenuity Systems, http://www.ingenuity.com/). Fisher’s exact test (*p* < 0.05) was used in IPA to estimate the probability that a specific set of proteins was related to a pathway.

### 4.5. Enzyme-Linked Immunosorbent Assay (ELISA)

CD55 protein was measured using a commercial quantikine ELISA kit (CSB-E05121h, CUSABIO, China) per the manufacturer’s instructions. Seventy cyst fluid samples–22 LGD, 5 HGD, 14 invasive IPMN, 13 MCN, and 16 SCN–were centrifuged to isolate the supernatant for the ELISA. Equal amounts of proteins (298 µg, as measured by BCA assay) were loaded into each well of a 96-well plate. The protein concentration data were analyzed statistically by student′s *t*-test.

## 5. Conclusions

In summary, we have generated the largest proteomic dataset of pancreatic cyst fluid to date and discovered potential markers of IPMN dysplasia using cyst fluid from 3 major types of PCLs (IPMN, MCN, and SCN) by LC-MS/MS. We significantly increased the protein coverage of each sample with a peptide library and discovered markers from a larger pool of candidates. By bioinformatics analyses, the DEPs were associated with biological functions that were related to pancreatic cancer, malignancy, and molecular secretion. Our process for discovering potential markers of IPMN dysplasia was logically sound. The agreement in the expression pattern of CD55 between the MS and ELISA data demonstrates that we have discovered reliable marker candidates of IPMN dysplasia. The development of cyst fluid markers can facilitate an accurate assessment of the degree of IPMN dysplasia and effectively guide surgical decision-making. Ultimately, if the developed marker is implemented in clinical practice, the accurate assessment of IPMN dysplasia will prevent unnecessary surgical resection for low-risk IPMN patients.

## Figures and Tables

**Figure 1 cancers-12-02383-f001:**
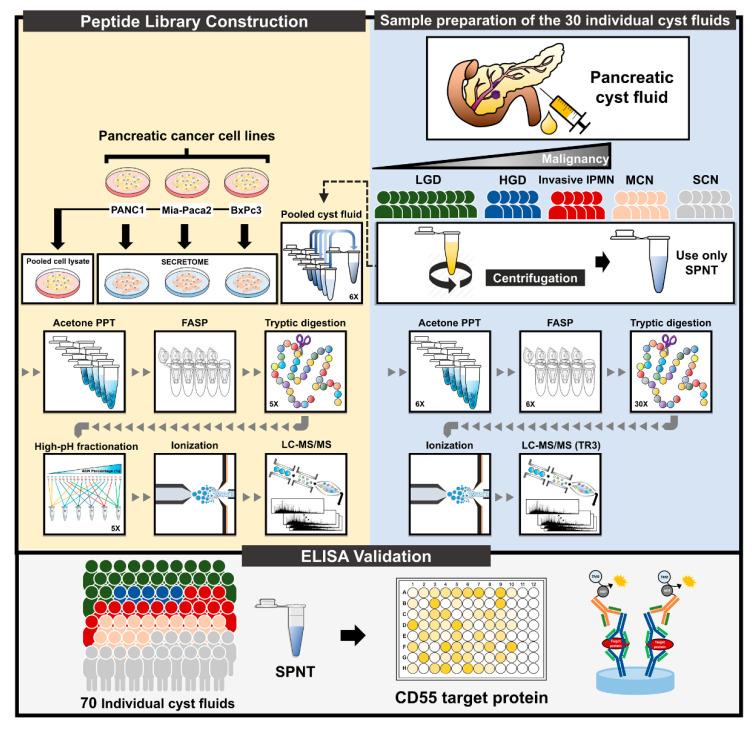
Experimental workflow. The overall experimental workflow comprises 3 sections: (1) Preparation of 30 individual samples, (2) peptide library construction, and (3) validation by ELISA. The cohort for label-free quantification included 30 pancreatic cyst fluid samples (10 LGD, 5 HGD, 5 invasive IPMN, 5 MCN, and 5 SCN). After mucus removal by sonication, the samples were centrifuged to isolate supernatant. Pooled cyst fluid (comprising equal amounts of 30 individual samples), secreted proteins from PANC1, Mia Paca-2, BxPC3, and pooled cell lysates from the 3 cell lines were compiled to generate a peptide library. All samples were precipitated using cold acetone to extract the protein. After FASP digestion, only the samples that were used to construct the peptide library were subjected to high-pH reverse-phase peptide fractionation. All peptides were analyzed on a Q Exactive mass spectrometer. CD55, one of the potential markers of IPMN dysplasia, was validated by ELISA. PPT, precipitation; FASP, filter-aided sample preparation; LGD, low-grade dysplasia; HGD, high-grade dysplasia; MCN, mucinous cystic neoplasm; SCN, serous cystic neoplasm; SPNT, supernatant.

**Figure 2 cancers-12-02383-f002:**
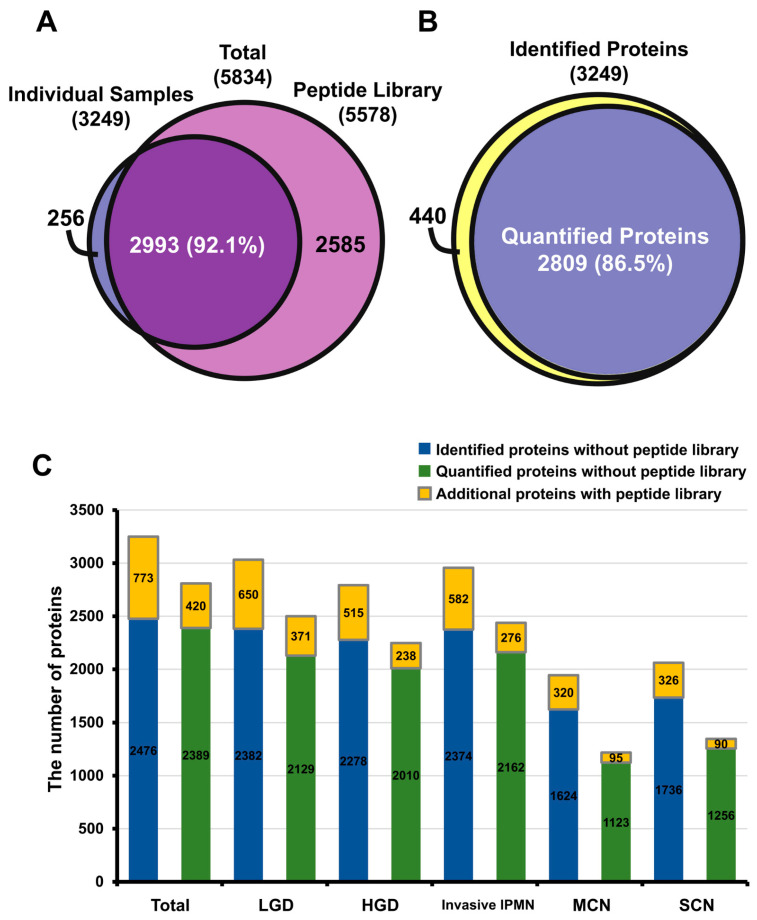
Comparison of protein identification and quantification in total dataset and in sample groups. (**A**) Among the 5834 proteins in the dataset, 5578 and 3249 were identified in the peptide library and individual samples, respectively, and 92.1% of the proteins overlapped. (**B**) Identified and quantified proteins in 30 individual cyst fluids; 86.5% of proteins were quantifiable. (**C**) Bar graph of the extra coverage enabled by the peptide library in the identified and quantified proteins of each sample group.

**Figure 3 cancers-12-02383-f003:**
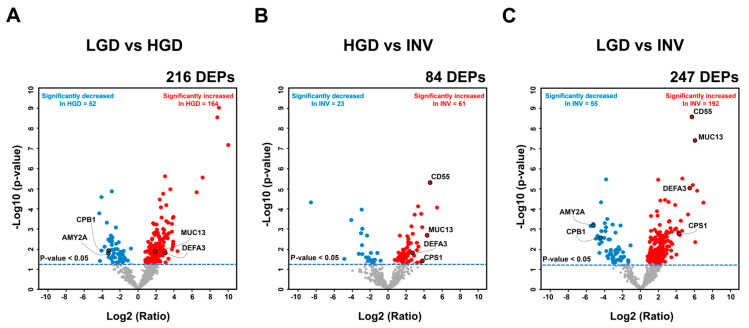
Volcano plots of differentially expressed proteins in three comparison groups. Student’s *t*-test (*p* < 0.05) was conducted for comparisons 1 (low-grade dysplasia (LGD) versus high-grade dysplasia (HGD)) (**A**), 2 (HGD versus invasive IPMN) (**B**), and 3 (LGD versus invasive intraductal papillary mucinous neoplasm (IPMN)) (**C**) to discover differentially expressed proteins (DEPs). The DEPs that were significantly expressed in each comparison group are indicated as colored dots (red: upregulated DEPs, blue: downregulated DEPs). Several marker candidates, including the validation target CD55, are highlighted in each comparison. DEP, differentially expressed protein.

**Figure 4 cancers-12-02383-f004:**
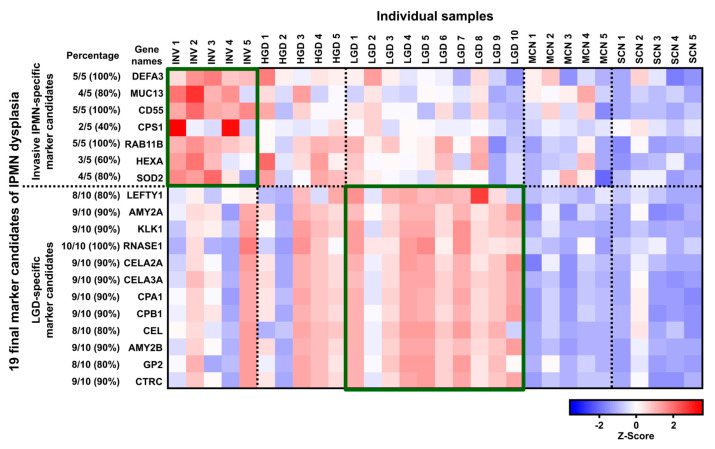
Overview of protein expression of the 19 final marker candidates of IPMN dysplasia. The heat map represents the expression patterns of the 19 final potential markers of IPMN dysplasia, based on z-score. Z-scores were calculated by averaging the LFQ intensities of the 3 technical replicates of each biological replicate. Red and blue reflect positive and negative values, respectively. Seven invasive IPMN-specific marker candidates were highly expressed in invasive IPMN. In contrast, 12 LGD-specific marker candidates were predominantly expressed in LGD. The percentage reflects the proportion of the 5 invasive IPMN samples showed upregulation of the given protein. Similarly, the percentage of LGD-specific marker candidates represents the number of LGD samples that showed upregulation for the given protein.

**Figure 5 cancers-12-02383-f005:**
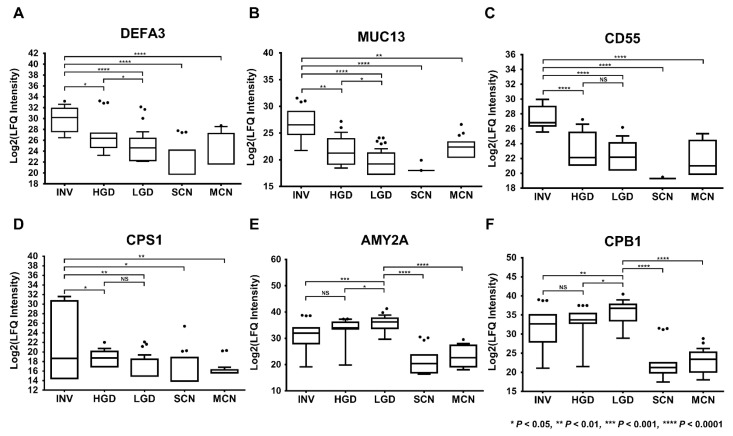
Six of the 19 potential markers that were differentially expressed in accordance with the degree of IPMN malignancy. DEFA3 (**A**), MUC13 (**B**), CD55 (**C**), CPS1 (**D**), AMY2A (**E**), and CPB1 (**F**) were differentially expressed, in accordance with the histological grades of IPMN. *, *p* < 0.05; **, *p* < 0.01; ***, *p* < 0.001; ****, *p* < 0.0001; NS, not available.

**Figure 6 cancers-12-02383-f006:**
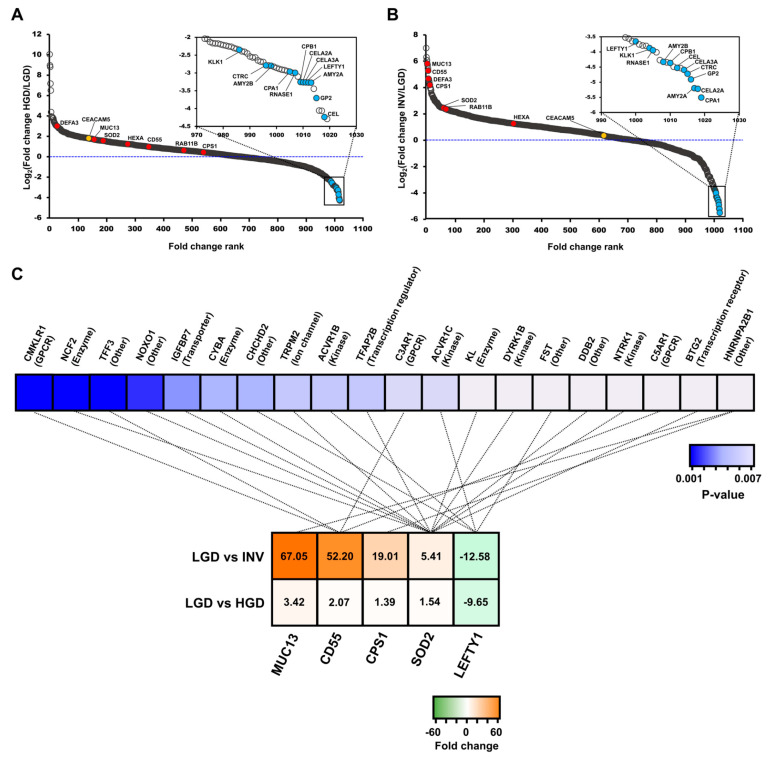
The dynamic range of protein fold-changes in comparisons 1 and 3 and the results of upstream regulator analysis in Ingenuity Pathway Analysis (IPA). The dynamic range with marked fold-changes of the 19 final marker candidates in comparisons 1 (LGD versus HGD) (**A**) and 3 (LGD versus invasive IPMN) (**B**). The red and blue dots indicate the log2-transformed fold-changes of the 7 upregulated and 12 downregulated proteins. The yellow dot represents the log2-transformed fold-change of CEA protein. (**C**) Five potential markers (MUC13, CD55, CPS1, SOD2, and LEFTY1) and their upstream regulators are connected by dotted lines. The molecular types of each upstream regulator are shown in parentheses. Each marker candidate is listed in order of decreasing fold-change, whereas the upstream regulators are listed in order of increasing p-values. LGD, low-grade dysplasia; HGD, high-grade dysplasia; INV, invasive IPMN.

**Figure 7 cancers-12-02383-f007:**
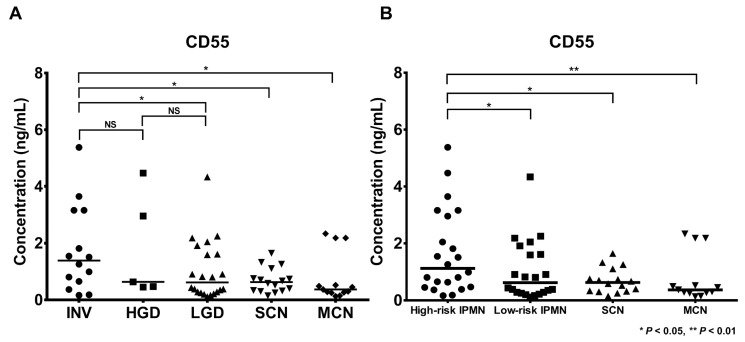
Validation of CD55 as a potential biomarker target by ELISA. ELISA was performed with CD55-specific antibodies to evaluate the validity of CD55 as a potential biomarker. The concentration patterns of CD55 by ELISA were generally consistent with the LFQ intensities. The CD55 concentrations are indicated according to two types of IPMN classification. The median CD55 concentrations were highest in invasive IPMN (**A**) and high-risk IPMN (**B**). LGD, low-grade dysplasia; HGD, high-grade dysplasia; INV, invasive IPMN; MCN, mucinous cystic neoplasm; SCN, serous cystic neoplasm; *, *p* < 0.05; **, *p* < 0.01; NS, not available.

**Table 1 cancers-12-02383-t001:** Demographic and clinical characteristics of the study population in the label-free quantification.

Characteristic	Pancreatic Cyst Fluids				
	Low-Risk IPMN	High-Risk IPMN		Other Cystic Lesions	
Group	LGD	HGD	Invasive IPMN	MCN	SCN
	(*n* = 10)	(*n* = 5)	(*n* = 5)	(*n* = 5)	(*n* = 5)
**Age (years)**					
	65.80 ± 5.55	67.80 ± 9.88	50.80 ± 14.45	49.00 ± 11.60	51.60 ± 17.08
**Gender**					
**Male**	5	4	3	1	1
**Female**	5	1	2	4	4
**Gland Type**					
**Gastric**	9	2 (1)	2 (1)		
**Intestinal**	0	1 (1)	1		
**Oncocytic**	0	0	1		
**Pancreatobiliary**	0	1	0		
**Pancreatic**	0	0	(1)		
**Unknown**	1	0	0		
**Duct Type**					
**Main**	0	0	1		
**Branch**	4	4	1		
**Mixed**	4	1	3		
**Unknown**	2	0	0		
**Cyst Focality**					
**Single**	8	5	4	2	0
**Multiple**	2	0	1	0	0
**Unknown**	0	0	0	3	5
**Mural Nodule**					
**Y**	2	3	5	0	0
**N**	8	2	0	2	0
**Unknown**	0	0	0	3	5
**Cyst Location**					
**Head**	4	3	3	0	1
**Body/Tail**	6	2	2	5	4
**Mixed**	0	0	0	0	0
**CEA Concentration (mg/L)**	1.44 ± 0.74	1.52 ± 1.04	5.48 ± 6.82	1.32 ± 0.96	1.44 ± 0.48
**CA 19-9 Concentration (mg/L)**	11.86 ± 8.69	22.80 ± 29.77	90.28 ± 129.71	20.20 ± 30.35	19.76 ± 17.81
**Cyst Size**					
**Cyst Size**	3.36 ± 1.33	3.56 ± 1.66	5.74 ± 3.69	7.50 ± 2.18	3.98 ± 1.93
**<3.0 cm**	4	3	2	0	1
**≥3.0 cm**	6	2	3	5	4

LGD, low-grade dysplasia; HGD, high-grade dysplasia; MCN, mucinous cystic neoplasm; SCN, serous cystic neoplasm; * The number of patients with two different gland types is shown in parentheses.

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
