# Peer review of "Marker Identification of the Grade of Dysplasia of Intraductal Papillary Mucinous Neoplasm in Pancreatic Cyst Fluid by Quantitative Proteomic Profiling"

_cancers, 2020, doi:10.3390/cancers12092383_

Round 1
Reviewer 1 Report
The authors have clarified several of the questions I raised in my previous review. Most of the major problems have been addressed by this revision. No further comments from this reviewer.
Reviewer 2 Report
The revised manuscripts showed substantial improvement over the last version. The Neutrophil staining and CD55 western blotting add the soundness and novelty to the manuscript. I therefore agree the manuscript to be published.
This manuscript is a resubmission of an earlier submission. The following is a list of the peer review reports and author responses from that submission.
Round 1
Reviewer 1 Report
Thank you for the opportunity to revise this paper. I found the topic very relevant and the paper well written and designed. I would to congratulate with the authors for this attempt, even if it is hard to imagine a direct application in clinical practice today. However, it represents an first step in the search of novel methods to distinguish the cystic pancreatic lesions in order to guide the treatment.
Reviewer 2 Report
Misol Do et al. report about intraductal papillary mucinous neoplasm (IPMN), using several
bioinformatics tools, they identified 19 final candidates consistently increased or decreased with greater IPMN malignancy. Finally, CD55 was validated in an independent cohort of 70 individual samples by ELISA. The manuscript deal with an interesting topic. Nonetheless, a few points should be addressed before considering the paper for publication.
- The authors employed using a commercial ELISA kit (CSB-E05121h, CUSABIO, China) in order to quantify the level of the CD55 marker. It is not specified if the tool is a DuoSet o Quantikine based kit and, importantly, it is not clear if the authors generated data supporting the precision of the assay. In other words, sensitivity, precision (repeatability), accuracy, matrix effects, and selectivity of these tools are of paramount relevance in evaluating the accountability of the proposed methods. Nonetheless, they did not employ the ROC method, therefore this suggestion might improve and increase the methodologic accountability.
A suggestion might be to assess the precision of the assay performing a set of evaluation comprising a report showing:
-values as the average of the coefficient of variation and their standard deviation of a variety of positive controls and negative controls;
-intraplate repeatability can be calculated by running a given number of replicates of each of control sample on 1 plate;
-interplate repeatability should be calculated by operating a given number of plates per 1 run.
Finally, between-run assessments should comprise run with corresponding controls.
- The authors make the statement to have novel candidates that represent potential biomarkers for IPMN dysplasia. Did the author try to develop uni- and/or multivariate risk model in order to predict the malignant evolution related to IPMN dysplasia in light of their findings? This is more than appropriate if the hazards proportionality of the variables included in the model can be assumed. Did the authors check this assumption? These are important information and, even if beyond the scope of this manuscript, should be explicitly mentioned in the paper, at least in terms of discussion in order to highlight strengths or study limitations by adding a brief comment to the already pinpointed statements (line 367-384, discussion section).
- Do original clinical and preclinical data exist about the translational relevance of the mentioned results? If yes, these elements should be presented, at least in the form of discussion and / or additional background from a short literature analysis. The authors themselves highlight the discrepancy between guidelines, and nomograms developed to predict low-risk and high-risk IPMN. One suggestion might be the role of inflammation, cytokine and microenvironment in mediating IPMN and the related prognostic impact, given the likelihood of pancreatic ductal adenocarcinoma (PDAC) evolution. Indeed, inflammation and the immune response are already extensively investigated in IPMN/PDAC, but its biological role remains partially obscure. Those are fundamental information in order to deeper validate the pieces of evidence discussed in the manuscripts. Porcelli et al (PMID: 30866547), Michael Goggins et al. (PMID: 12651607) and other authors comprehensively summarized the mechanism underlying the aggressive cancer phenotype acquisition from the cancer microenvironment and inflammation perspectives. Moreover, the proposed data nicely match with the novel insights derived from the extensive bioinformatic analysis performed by the authors that partially parallels the above-mention and other’s evidence regarding highly expressed genes previously reported in pancreatic cancer and confirmed by significance analysis of microarray and in vitro/ex vivo models (i.e. CD55 and TGF-beta, PMID: 30866547). The authors should provide insights in this regard, with a particular biological focus discussing tumour-associated microenvironment cells (such as soluble mediators, a cytokine produced by macrophages and cancer-associated fibroblasts) role in aggressive phenotype and drug resistance acquisition, especially in the discussion, where two seminar manuscript is mentioned (lines 325-27 pancreatic cyst fluid contains secreted proteins from surrounding tumour cells [32, 33, 49] and plasma proteins that penetrate into the cyst epithelium due to tissue injury or the enhanced permeability and retention (EPR) effect of the surrounding blood vessels [53]), alluding to tumour immune-microenvironment. This reviewer personally misses some potential translational relevance that takes into account the tight relationship existing between the tolerogenic immune infiltrate and the potential neoplastic evolution (PMID: 31277479; PMID: 20955708). Those references would significantly increase the relevance of the manuscript contents for the oncologic community.
Linguistic revision by a native speaker can be recommended.
Reviewer 3 Report
Do et al describe the identification of potential protein markers for different grades of IPMN in the pancreatic cyst fluid using MS and Elisa. Their findings are very interesting and could potentially benefit the grading of IPMN and other dysplasia. However, I have several minor concerns which the authors need to address before I would be fully convinced.
- The title claims to be able to “predict” the grade of IPMN, but in the paper, no prediction was made as authors were using biological samples of known grade. It would be more appropriate to claim something like “marker identification” instead of “predicting” in the title.
- The IPMN marker DEAFA3 and Cd55 are neutrophil related protein. It would be interesting to look at whether there is neutrophil enrichment in the pancreatic cyst fluid of invasive IPMN or in the IPMN tissue sections. It would also be interesting to perform immunostaining of CD55 in invasive IPMN tissue sections to identify what types of cells shows elevated expression of CD55.
- The ELISA data of CD55 has lower statistical significance compared with MS data. Since ELISA needs two antibodies binding to the sample protein for identification, and pancreas is a protease-rich tissue, I was wondering whether the lower statistical significance was due to degradation of the CD55 protein by proteases secreted by the pancreas. I would suggest the authors to perform Western blot to see if CD55 has a higher expression level in invasive IPMN pancreas cyst fluid.
- Please in the discussion section include a paragraph discussing the GL and KEGG finding. Describe the biological relevance of the few most enriched pathways (eg. The complement pathway enrichment in KEGG).
- Please cite the following papers as they are relevant to your studies.
Hata, Tatsuo, et al. "Predicting the grade of dysplasia of pancreatic cystic neoplasms using cyst fluid DNA methylation markers." Clinical Cancer Research 23.14 (2017): 3935-3944.
Hata, Tatsuo, et al. "Cyst fluid telomerase activity predicts the histologic grade of cystic neoplasms of the pancreas." Clinical Cancer Research 22.20 (2016): 5141-5151.
Park, Jisook, et al. "Discovery and validation of biomarkers that distinguish mucinous and nonmucinous pancreatic cysts." Cancer research 75.16 (2015): 3227-3235.